# Advancing Diabetic Retinopathy Research: Analysis of the Neurovascular Unit in Zebrafish

**DOI:** 10.3390/cells10061313

**Published:** 2021-05-25

**Authors:** Chiara Simone Middel, Hans-Peter Hammes, Jens Kroll

**Affiliations:** 1Department of Vascular Biology and Tumor Angiogenesis, European Center for Angioscience, Medical Faculty Mannheim, Heidelberg University, 68167 Mannheim, Germany; chiara.middel@medma.uni-heidelberg.de; 2Fifth Medical Department and European Center for Angioscience, Medical Faculty Mannheim, Heidelberg University, 68167 Mannheim, Germany; hans-peter.hammes@medma.uni-heidelberg.de

**Keywords:** diabetic retinopathy, zebrafish, neurovascular unit, microvascular complications and dysfunction, metabolism

## Abstract

Diabetic retinopathy is one of the most important microvascular complications associated with diabetes mellitus, and a leading cause of vision loss or blindness worldwide. Hyperglycaemic conditions disrupt microvascular integrity at the level of the neurovascular unit. In recent years, zebrafish *(Danio rerio)* have come into focus as a model organism for various metabolic diseases such as diabetes. In both mammals and vertebrates, the anatomy and the function of the retina and the neurovascular unit have been highly conserved. In this review, we focus on the advances that have been made through studying pathologies associated with retinopathy in zebrafish models of diabetes. We discuss the different cell types that form the neurovascular unit, their role in diabetic retinopathy and how to study them in zebrafish. We then present new insights gained through zebrafish studies. The advantages of using zebrafish for diabetic retinopathy are summarised, including the fact that the zebrafish has, so far, provided the only animal model in which hyperglycaemia-induced retinal angiogenesis can be observed. Based on currently available data, we propose potential investigations that could advance the field further.

## 1. Introduction

Diabetes mellitus is one of the most prevalent metabolic conditions worldwide. The International Diabetes Foundation (IDF) estimated in 2015 that there were 415 million adults aged 20–79 living with diabetes. Due to increasing populations and the high prevalence of obesity in developed countries, this number is expected to rise to 642 million people by 2040 [1].

Diabetic retinopathy (DR) is a frequent microvascular complication occurring in patients with both type 1 or type 2 diabetes, and remains a leading cause of vision loss and blindness globally [2]. The probability of developing DR is highly dependent on the duration of diabetes and the level of glycaemic control. Furthermore, the management of other risk factors such as hypertension can also have a significant effect on the development of DR [3].

Due to earlier detection and improved treatment options, the prevalence of both retinopathy and sight-threatening stages has declined in recent years [3,4]. However, since the number of patients with diabetes and the average lifespan will increase globally in the coming decades, DR will continue to be a highly relevant condition in the foreseeable future [5,6].

The clinical aspects of DR have been thoroughly characterized [2,7,8]. There are several changes in the retinal vasculature that can be attributed to hyperglycaemia such as pericyte and endothelial cell loss. These are accompanied by altered blood flow and altered vascular permeability. The first ophthalmologically visible signs of DR are microaneurysms and haemorrhage, followed by hard exudates—the cardinal signs of non-proliferative DR (NPDR). In moderate stages, additional vascular abnormalities follow, such as the important intraretinal neovascularization. In later stages, due to increasing ischemia, retinal neovascularization extends through the inner limiting membrane (ILM) and along the surface of the retina or into the vitreous cavity. This stage is referred to as proliferative diabetic retinopathy (PDR). The complications associated with PDR, such as vitreous haemorrhage, retinal detachment or macular nonperfusion, may lead to vision loss. A common denominator in these complications is the associated photoreceptor dysfunction. 

The UK Prospective Diabetes Study (UKPDS) showed that, at the turn of the century, 38% of newly diagnosed patients with type 2 diabetes already showed some stage of retinopathy [9]. In a large European population-based study published in 2016, 21% of patients with screening-detected type 2 diabetes already showed signs of DR [10]. This indicates that, while diabetes development has been overlooked in patients in the past, there has been a decline in this failure. A deeper understanding of the complex pathophysiology underlying the early stages of diabetic retinopathy is needed to develop new concepts for personalized medicine.

Zebrafish (*Danio rerio*) have been used as a model for human disease for decades. Their ease of maintenance and relatively short reproduction time make them a very attractive model organism. Their small size, fast development and their ability to produce up to 200 offspring per week underscore these advantages. Adult zebrafish measure 3 to 5 cm in length and can be housed in tanks with up to 30 other fish, depending on the tank size. Embryogenesis is almost complete, and most organs are developed at 72 h post fertilisation (hpf). Zebrafish are considered adults at 3 months of age. Since the larval development happens outside of the mother and the larvae are transparent, development can be closely monitored in vivo. Additionally, zebrafish show a high degree of genetic, anatomical and physiological similarities to humans [11,12,13].

In recent years, zebrafish have increasingly been used to investigate diabetes and other metabolic disorders. [14] Inducing diabetes in zebrafish can be performed in various ways. Diabetes can be induced through classical external approaches, such as injection of streptozotocin (STZ) [15]. Another approach in zebrafish is immersion in high-glucose solutions [16]. However, since the genome-wide association studies (GWAS) systematically identified various genetic loci that are associated with different kinds of disorders, including type 2 diabetes mellitus and obesity [17,18], zebrafish, as a well-established animal model for forward and reverse genetic methods, have gained attention in the field [19,20].

In this review, we discuss numerous pathologies associated with DR that have so far been identified in zebrafish models of diabetes and offer an overview of experimental techniques and perspectives for future investigators in the field.

## 2. Zebrafish in Diabetic Retinopathy Research 

To investigate pathologies associated with DR in zebrafish, it is important to examine the various cell types that are implicated in the development of DR and to recognize differences between mammalian and zebrafish models of DR. 

### 2.1. The Neurovascular Unit in Mammals and in Zebrafish—Similarities and Dissimilarities

The term neurovascular unit was first applied to the blood–brain barrier and, later, to the inner blood-retinal barrier. It defines the functional and structural coupling of vascular cells, i.e., endothelial and vascular mural cells (especially pericytes), neural cells, which encompass ganglion cells, amacrine cells, horizontal cells and bipolar cells as well as macro- and microglia [21]. These cell types work closely together to regulate the nutrient and oxygen levels in the functional retina through regulation of blood flow and trans- and paracellular transport. Several reviews have discussed this complex cellular crosstalk and its disruption by diabetes on various levels [22,23,24,25]. 

Zebrafish have long been accepted as a valuable model to study eye disease [26,27]. Both the anatomy and the function of the retina and the NVU have been highly conserved in vertebrates. The mammalian and the zebrafish retina both consist of three nuclear layers and two synaptic (plexiform) layers, and contain the same cell types (Figure 1). In zebrafish as well as in mammals, the phototransduction cascade is activated when light reaches the photoreceptors. They transport the information to the bipolar cells in the outer nuclear layer (ONL), with their synapses interacting in the outer plexiform layer (OPL). In the OPL, horizontal cells, which are local interneurons, regulate the photoreceptor output. The bipolar cells, which have their cell bodies in the inner nuclear layer (INL), activate retinal ganglion cells via synapses in the inner plexiform layer (IPL). This interaction is modulated by amacrine cells. The axons of the ganglion cells merge on the vitreous surface of the retina and form the optic nerve, through which they leave the retina and reach the visual cortex of the brain [28,29]. One key difference between the zebrafish retina and the mammalian retina is that the zebrafish retina can regenerate, a phenomenon discussed further in the chapter on immune cells. 

Data describing the nature of the blood–retinal barrier in zebrafish are scarce. There are numerous studies confirming that the blood–brain barrier in zebrafish is highly conserved [30,31]. The main components of the NVU can be found in zebrafish, including a single, continuous endothelial cell layer with tight junctions to control paracellular transport, pericytes that cover the abluminal vessel wall and are covered by a basal lamina and radial glia processes that are in permanent contact with both the endothelial cells and neural cells to ensure proper vascular function.

### 2.2. Endothelial Cells

Under physiological conditions, a single layer of non-fenestrated endothelial cells forms the luminal wall of vessels and ensures vessel integrity through communication with the surrounding cells. Especially in the brain, and by extension in the retina, the endothelium is very restrictive in order to protect the neurons from toxins and metabolites. It ensures this selective permeability through inter-endothelial cell junctions such as tight junctions, adherence junctions and gap junctions, as well as tightly controlled transcellular pathways. Many molecules have been identified that play an important role in retinal endothelial junctions [32].

Critical features of DR include vascular dysfunction, which is associated with increased vascular permeability because of the loss of tight junctions, and loss of endothelial integrity [24]. This leads to hypoxia in the poorly perfused retina that induces an increase in levels of Angiopoietin-2 (Ang-2) and vascular endothelial growth factor (VEGF). Increased levels of VEGF, in turn, lead to the formation of new and more fragile blood vessels in the retina. This pathological neovascularization is a key component of irreversible causes of blindness in various retinopathies, as it leads to bleeding into the vitreous and retinal detachment due to macular oedema. Upregulation of VEGF is an important example of how the knowledge of pathological pathways can lead to the development of new treatments. An intravitreal injection of anti-VEGF antibodies is the only working treatment for advanced stages of retinal neovascularization. The discovery of the molecular pathways associated with the upregulation of VEGF, and the subsequent development of a new treatment option, show how crucial adequate animal models are for the development of new treatment methods [33,34].

Zebrafish have distinct advantages regarding their retinal vasculature because there is an extensive availability of reporter lines, e.g., the *Tg(fli1a:EGFP)* line, in which the whole endothelium is visualized due to the expression of enhanced green fluorescence protein (EGFP) [35]. Therefore, most of the research that has been performed on zebrafish in the context of diabetes has been focused on vascular pathologies. Quantification of endothelial cells and the whole retinal vasculature under different conditions is facilitated by the use of the appropriate reporter line. Reporter lines for other cell types are discussed below.

The development of zebrafish retinal vasculature has been well studied [36] and reviewed [37,38,39]. Therefore, we will only give a short overview of the development of the vessels and the key differences in comparison to mammals. The blood supply in the human retina and in some rodents is provided by two vascular plexuses, the choroid and the intraretinal vasculature. During development, the retinal vascular network undergoes intense remodelling. Blood supply during early development is provided by the hyaloid vasculature. Once the primary plexus of the retinal vasculature starts to grow into the retina, the hyaloid vasculature regresses. In humans, the growth of the primary plexus is controlled by astrocytes through the secretion of VEGF. The intraretinal vasculature grows from the primary plexus through angiogenesis, as the development of the retinal vasculature continues [40,41]. In humans, the hyaloid vasculature forms during late embryogenesis and the switch to retinal vasculature starts mid-gestation while, in mice, the switch happens at birth.

One major difference between mammalian and zebrafish vasculature is that zebrafish vasculature does not exhibit this switch in a vessel origin. The first endothelial cells are present by 48 hpf and are localised between the lens and the retina. They give rise to the hyaloid vasculature, which at first adheres tightly to the lens. By 5 days post fertilisation (dpf), the hyaloid vasculature enwraps the lens entirely and forms the peripheral circumferential vein, the inner optic circle (IOC). From 15 dpf on, they lose contact with the lens and become more and more attached to the retinal surface. There is no further invasion of vessels. Zebrafish only have vessels on the surface of the retina and on top of the inner limiting membrane (ILM). This suggests that, comparable to some mammalians such as the guinea pig, the relatively thin retina does not need an intraretinal plexus because oxygen supply can be ensured solely through diffusion [36]. This is another major difference between mammalian and zebrafish retinas, one which has consequences when analysing the vasculature of diabetic phenotypes. To visualize retinal vessels in mammals properly, the ILM needs to be removed during dissection. If the ILM is removed in zebrafish, this would also remove the retinal vasculature. This difference is essential in the context of DR research: the new blood vessels that develop in PDR break through the ILM. This is not possible in zebrafish.

#### 2.2.1. Non-Genetic Zebrafish Models of Diabetic Retinopathy 

Zebrafish have been used to model pathologies associated with DR for well over a decade now. The first approach was to induce hyperglycaemia in zebrafish through immersion in glucose solutions (Table 1). This was achieved by placing them in alternating solutions of 2% and 0% glucose every 24 h for up to 30 days, showing a decrease in the thickness of the IPL and the INL in the retina [16], which has also been observed in other animal models and diabetic patients through spectral domain optical coherence tomography [42]. This strengthens the position of zebrafish as a model to study the effects of high glucose levels on the retina, since the short duration of high glucose exposure, in combination with the ease of vascular visualization and the large breeding size, allow for convenient experimentation.

A re-evaluation of the model found various signs of retinopathy, apart from neurodegeneration. After a 30-day incubation period in alternating 2% and 0% glucose solutions, treated fish exhibited an increase in vessel diameter and a thickening of the vessel basement membrane, as well as prominent defects in cone photoreceptors with signs of photoreceptor degeneration, including impaired electroretinography (ERG) results. Other vascular pathologies were visible as well, such as wider tight and adherent junctions that suggest increased vessel permeability and upregulation of VEGF; however, those were visible in mannitol-treated control fish as well, which led the authors to reconcile these changes with hyperosmolarity rather than high glucose [43].

Since then, diabetes-like metabolic conditions have been induced through different methods such as a zebrafish model of experimental hypoxia. Hypoxia was achieved through a device that perfused N_2_ gas directly into the water inside a sealed aquarium, preventing air from leaking out. The system automatically maintained a constant level of O_2_ in the water and thus placed the zebrafish in 10% air saturation. The authors could show that 12 days of hypoxia treatment increased the density of capillary networks significantly, and that hypoxia, therefore, had an angiogenic effect in the zebrafish retina. They also established a dose-dependent relation of hypoxia to angiogenesis through exposure of zebrafish to different concentrations of air-saturated water and analyses of the different angiogenic responses. Most importantly, this neovascularization could be blocked by oral anti-VEGF agents (sunitinib and ZN323881), confirming that neovascularization in zebrafish is as VEGF-dependent as it is in mammals [44]. 

Experimental hypoxia is a strategy that has been used to model PDR in rodents, since diabetic mammalian models do not develop spontaneous neovascularization [45]. The best known rodent model is the oxygen-induced retinopathy (OIR) in mice, which has recently been reviewed, with all its advantages and disadvantages [46]. In rodent models, it is common to induce diabetes through STZ injection. STZ is an antibiotic that leads to sustained hyperglycaemia through the disruption of pancreatic islets of Langerhans and the destruction of beta-cells. Intraperitoneal or direct caudal fin injection of 300–350 mg/kg STZ in zebrafish leads to an increase in fasting blood glucose and, in addition, a marked decrease in retinal photoreceptor layer (PRL) and IPL thickness [47]. The administration of i.p. STZ-injections on days 1, 3 and 5, and subsequent booster injections once a week for two more weeks (day 12 and 19), can induce sustained hyperglycaemia for up to at least three weeks [47]. A later study, however, found that, when following the proposed protocol, the zebrafish showed a high mortality and increased levels of hypoglycaemia—indicators that this is an imperfect method of inducing diabetes in zebrafish [48].

In short, the most common ways of inducing diabetic metabolic states or retinal neovascularization in mammalian models of retinopathy (glucose exposure, hypoxia and STZ injections) can be used in adult zebrafish, and lead to pathologies that show similarities with DR in humans. Variants of the methods presented in this chapter have been used frequently, especially the high-glucose model, to study the effects of new potential anti-angiogenic drugs.

#### 2.2.2. Genetic Zebrafish Models of Diabetic Retinopathy 

The main advantage of zebrafish in diabetes research is the ease of genetic manipulation. The possibility of introducing targeted mutations using sequence-specific transcription activators such as effector nucleases (TALENs), or the clustered regularly interspaced short palindromic repeats (CRISPR) system, have made the zebrafish highly attractive for studying the consequences of loss-of-function alleles in an effective way (Table 1) [49,50].

After the successful establishment of the Zebrafish Mutation Project [51], the effects of the null mutation of *pdx1* (pancreatic and duodenal homeobox 1) in zebrafish were analysed. The analysis indicates that homozygous mutation leads to impaired pancreatic islet development and disrupted glucose homeostasis [52]. Subsequent analyses by the same group showed that these mutants exhibit distinct signs of retinal vasculature dysfunction, including vessel constriction, points of stenosis, a reduction of average vessel diameter, tortuous vessels with increased vessel density and increased sprouting and branching, as well as a reduced expression of ZO-1 (zonula occludens protein 1). ZO-1 is a molecule that is integral to tight junctions which are responsible for connecting endothelial cells and regulating permeability in vessels. Furthermore, GLUT1 expression was largely absent in mutants compared to wildtype controls [53]. Changes in GLUT1 expression have also been reported in DR patients and mouse models [54]. Parallel research on a CRISPR/Cas9-induced homozygous *pdx1* mutant by our group independently observed the same findings of hypersprouting and hyperbranching in the adult retina. This study additionally described that a pharmacological modulation of VEGF and Nitric Oxide signalling rescues the hyperglycaemia-induced changes in the vasculature [55]. Recently, work on a homozygous *aldh3a1* knockout zebrafish line showed a moderate retinal vasodilatory phenotype, which could be aggravated through experimental diabetic conditions achieved through *pdx1* expression silencing. This study provides evidence that 4-Hydroxynonenal (4-HNE), which has been implicated as a clinical feature in patients with diabetes and diabetic complications before, induces impaired glucose homeostasis and causes retinal vascular alterations [56]. Another interesting genetic model of hyperglycaemia associated retinal pathologies is the combination of a CRISPR/Cas9-induced mutation in the *glo1* (Glyoxalase 1) gene with an overfeeding protocol, which includes an overfeeding period of 8 weeks with artemia [57]. Glyoxalase 1 is an enzyme which catabolizes methylglyoxal (MG), a reactive metabolite that is a main precursor to advanced glycation end products (AGEs), and is elevated in the plasma and tissue of diabetic patients. Loss of glyoxalase 1 can therefore lead to increased levels of MG and a diabetic phenotype [58]. This protocol also leads to an increased angiogenic sprout formation of the retinal vasculature [57], and was recently scored as the best type 2 diabetes model in zebrafish that is directly followed by both *pdx1* mutants [59].

These findings are exciting from multiple perspectives. The confirmation that two different *pdx1* mutants by two independent groups produce a phenotype resembling DR in zebrafish establishes this zebrafish mutant line as a reliable and reproducible model for further research into the mechanisms associated with DR, particularly retinal angiogenesis in adults [53,55]. Furthermore, the homozygous *pdx1* mutant is the first animal model to show retinal angiogenesis under hyperglycaemia. Additionally, these results show that genes potentially involved in the pathogenesis of diabetes can be tested in zebrafish to find out whether and in which dimension they are involved in the development of DR. This could catalyse research into potential pathways and treatments, such as high-throughput screenings for potential targets of personalized treatment methods.

#### 2.2.3. Small Molecule Testing on Zebrafish Larvae

The main principle of the ethical use of animal studies in clinical research is that of the 3 R’s: researchers should always work to replace, reduce and refine animal studies. Zebrafish have increasingly come into focus to achieve these goals. 

In the context of DR, this especially applies to small molecules that are supposed to stop or delay neovascularization. Currently, the gold standard for treating sight-threatening PDR is pan-retinal laser coagulation, supplemented by intravitreal injections of anti-VEGF agents when clinically significant macular oedema (CSME) is present. Clinicians need alternative therapies targeting other pathways and potentially allowing for non-invasive application methods. Using zebrafish larvae to study the effects of antiangiogenic drugs can replace some animal toxicity studies, since zebrafish larvae start being considered independent organisms at 5 dpf. Studying the effect of new agents on the developing zebrafish hyaloid vasculature can screen them for their function and thus leave a smaller number of potential pharmaceutical agents which have already proven their efficiency in zebrafish larvae to be tested on mammals [60,61]. 

A retinal phenotype in the *pdx1* zebrafish line has been observed, even in the larval hyaloid vasculature. The phenotype could be rescued through incubation in metformin and PK11195, providing evidence that the phenotype is caused by hyperglycaemia [55]. This highlights the suitability of the model: shortly after the induction of the mutation, there is already in vivo imaging available to confirm whether the changes in the retinal vasculature can be rescued through the application of a specific drug. 

Wildtype zebrafish larvae that are incubated in glucose for 3 days, starting at 3 dpf, show an increased diameter in hyaloid vessels and upregulated expression of VEGF RNA [62]. This can easily be used to evaluate the efficacy of an antiangiogenic drug. However, screening for antiangiogenic agents does not necessarily require a model associated with hyperglycaemia [63]. The physiological development of the hyaloid vessels can also be disrupted by anti-angiogenic agents, and this disruption can be observed in vivo in a matter of days. As already proposed in 2003, zebrafish, with their rapid development and optical transparency, are exceptionally convenient for high-throughput in vivo screening of anti-angiogenic agents [64]. Their use in such experiments has been on the rise for years [65,66,67] because researchers want to develop treatments to inhibit neovascularization as the main reason for vision loss in various ocular diseases including, but not limited to, DR. Most recently, a protocol for drug pooling has been established. In this study, the authors evaluated the usefulness of pooling various agents and incubating zebrafish larvae with multiple agents at a time [68]. This enhances and facilitates the screening of ocular anti-angiogenic drugs in zebrafish larvae, making it possible to use even fewer animals in a first line screening of novel agents. 

### 2.3. Pericytes

Pericytes are specialised mural cells (MCs) which occupy the abluminal side of the vessel wall and are in constant communication with endothelial cells, microglia and neurons. Their location and morphology are very distinct. They sit within the basal membrane with long processes covering the vessel walls [69]. Pericytes contribute to the regulation of blood flow through the retinal vessels and thereby to the oxygen supply for the retina, as well as the anatomical stabilization of the BRB [70]. Additionally, they are important for the formation of new vessels. As such, they are recruited for developing vessels through chemotactic factors such as platelet-derived growth factor B (PDGF-B), which is secreted by endothelial cells, and the interaction of Angiopoietin 1 (Ang1), which pericytes express, and the endothelial cell derived tyrosine-kinase receptor Tie-2. A lack of pericytes leads to severe endothelial dysfunction and even perinatal death in PDGF-B-deficient mice through the absence of functional blood vessels [71]. Angiopoietin 2 (Ang2), on the other hand, which is upregulated in patients with DR, has been found to disrupt the PDGF-B stimulated pathway and subsequently impair communication and the recruiting of pericytes to vessel walls. 

In the retina, the pericyte-to-endothelial-cell ratio is 1:1, which emphasizes the importance of their role in ensuring the structural and functional integrity of the vascular architecture and the BRB [69,72]. An early feature of DR is the depletion of capillary pericytes. Several mechanisms have been suggested to play a role in pericyte loss. These include a reduction in PDGF-B signalling due to hyperglycaemia and increased secretion of Ang2 by endothelial cells. Further mechanisms implicated in pericyte depletion are microglial pro-inflammatory-mediated activation of pro-apoptotic molecules, reactive oxygen species (ROS) damage to pericyte mitochondria by activating apoptotic cascades, induction of apoptosis through generation of tumour necrosis factor-alpha (TNF-alpha) and advanced glycation end-product (AGE) and glutamate excitotoxicity [73]. Pericyte loss contributes to the eventual formation of acellular capillaries and microaneurysms, as well as haemorrhage, and consequently hypoperfusion in the retina. 

Pericytes are present in the retinal vasculature of zebrafish. It was confirmed through electron microscopic analysis of the ultrastructure of retinal vessels that zebrafish vitreo-retinal vessels carry mature pericytes that are located on top of the vascular endothelium within the basal lamina, which is the same location that they inhabit in mammalian vessels. This is observed both in young and senescent specimen but not in larvae [36]. Neither in mammals nor in zebrafish do pericytes express one specific cell marker [74], and not all markers can be used across different species. Of the most prominent and commonly used pericyte markers, pericytes in zebrafish express PDGF receptor beta (PDGFRβ) and Notch3 [75]. More recently, transgenic reporter lines have been developed for live imaging of MCs. In one study, EGFP, mCherry or the Gal4FF drivers are expressed under the control of the *pdgfr**β* promotor and indicate that the first *pdgfr*β positive cells, which are most likely pericytes, can be observed at the 8-somite stage in the cranial neural crests of zebrafish [76]. Two zebrafish smooth muscle actin (*sma*) homologues have been reported: *acta2* and *transgelin*. SMA is used to stain vascular smooth muscle cells (vSMCs) [77]. These can include pericytes, *acta2* and *transgelin*; however, they do not seem to be specific for pericytes. Specifically, one study found that the retinal vessels in the centre of the optic disc were arteries and densely covered by *transgelin1* and *acta2* positive cells. This suggests that those cells were vSMCs, while capillaries and venous vessels were covered with PDGFRβ positive cells. This confirmed the finding previously established in the mammalian retina, as mentioned above, that there is no one pan-pericyte marker [78]. These findings were in accordance with the observation that vSMCs are typically found in large blood vessels and are separated from the endothelium by the basement membrane, controlling vessel contractility and regulating blood flow. Pericytes as specialised MCs are rather found in microvessels, especially in the brain and the eye, and are embedded in the basement membrane. Even though both vSMCs and pericytes may come from the same cell line and express similar molecular markers at various time points, they are defined as two different cell types. [69] 

To study the development of vSMCs in vivo, reporter lines were established that express GFP or mCherry under the mural cell promotor *acta 2*. In these reporter lines, it was established that vascular mural cells turned on *acta2:EGFP* several days after the initiation of circulation, and were morphologically similar to pericytes in early development. The larger head vessels were associated with *acta2:EGFP* positive cells at 7 dpf. In this study, the authors suggested that zebrafish do not need as many mural cells as mammals, which thus explains the difference in expression of acta2 and the late association of mural cells with the vessels, as mammalian blood pressure is much higher than that in zebrafish. Accordingly, this could mean that zebrafish vessels do not need the stabilizing effect of mural cells as much as mammals do. [79] We have, however, observed that, at least in the retina, the ratio of one pericyte to one endothelial cell is most likely conserved in zebrafish.

Recently, to research the function of the *frizzled4* gene, which is implicated in the development of familial exudative vitreoretinopathy (FEVR), one study found that pericytes in the zebrafish retina contain *frizzled4* mRNA and have a very unique position on the retinal vasculature [80]. These observations could finally make the quantification of pericyte numbers in the zebrafish retina possible. To quantify pericyte numbers in rodent models of DR, or in diabetic donor eyes, the retinal digest method is used [81]. In Figure 2, we provide an image of a retinal trypsin digest of the zebrafish retina and indicate the different cell types (endothelial cells and pericytes) that can be visualized through this method.

### 2.4. Microglia

Microglia are the resident macrophages of the central nervous system (CNS) and an important part of the neurovascular unit. Their morphology is unique, with small cell bodies in the plexiform layers and long cell processes that may span all the nuclear layers. They monitor and control the surrounding microenvironment in the CNS, and they are able to synthesize and release multiple cytokines, chemokines, neurotrophic factors and neurotransmitters. Under physiological conditions, microglia receive inhibitory signals from the surrounding microenvironment, such as secretion of transforming growth factor beta (TGFβ) by the retinal pigment epithelium (RPE) and expression of CD200 on several retinal cells and the release of CX3CL1 by healthy neurons or endothelial cells. TGFβ even induces an anti-inflammatory effect [82]. 

The activation of microglia, noticeable through morphological transformation and migration of residential microglia, is a common hallmark sign of retinal disease. Microglia are involved in all stages of DR. In early stages, there is a moderate increase in perivascular microglia and they appear to be slightly hypertrophic. During non-proliferative stages of DR, microglia tend to cluster around lesions such as microaneurysms [83]. In proliferative stages, the new vessels, which are highly fragile and dilated, are surrounded by microglia [82,84]. If chronically activated, microglia play a part in neuroinflammation by constantly releasing cytokines, and thus lead to an increased invasion of immune cells into, and further damage to, the retina through inflammatory processes [84,85]. 

At present, there are no studies on microglia or inflammatory processes in zebrafish models of DR in general. There are various reporter lines in which microglia/macrophages express fluorescence proteins, as well as the possibility of staining microglia with antibodies against lymphocyte cytosolic *plastin 1* (*lcp1*, a pan-leukocyte maker) and *4C4* (which stains microglia exclusively in zebrafish) [86,87]. 

### 2.5. Müller Glia

Microglia are involved in a constant active crosstalk with Müller cells. Müller cells and astrocytes are the resident macroglia in the retina, with Müller cells being the major glial cell type in the mammalian retina. In all, 90% of retinal glia are Müller cells [84]. They are in intimate contact with all other cell types, with their end-feet in close contact with ganglion cells and endothelial cells on the vitreal side of the retina, and with photoreceptors on the outer side of the retina. Their cell bodies are found in the inner nuclear layer, but their processes span the entire retina. 

There are three main functions that are being attributed to Müller glia: the uptake and recycling of neurotransmitters and retinoic acid compounds, the control over the metabolism and the supply of nutrients to the retina, as well as the regulation of blood flow and maintenance of the BRB [88]. 

In diabetic conditions, Müller cells are a potential source of growth factors, e.g., VEGF, that are effectors of neovascularization, and cytokines, which lead to the activation and migration of microglia. After prolonged periods of overstimulation, Müller glia begin to die, leading to photoreceptor degeneration, vascular leakage and intraretinal neovascularization [84,89,90]. The first marker of activation of Müller glia is an increase in expression of glial fibrillary acidic protein (GFAP), which is the most common marker of gliosis.

Müller cells become activated in hyperglycaemic conditions in adult zebrafish, as shown by antibody staining against GFAP or glutamine synthetase (GS) [43,53,78]. However, while in rodent models and in human eyes, healthy Müller cells do not express GFAP, Müller glia in zebrafish always express GFAP, thus rendering the demonstration of Müller cell activation difficult. This may be caused by the zebrafish retina’s regeneration capability after injury, which has been reviewed elsewhere [91,92]. The source of regenerated neurons in zebrafish is the Müller cells themselves; upon detecting an injury, they re-enter the cell cycle and undergo asymmetric division, ultimately generating multipotent progenitors that replace the lost neurons [93]. This may pose an interesting addition to DR research in mammalian models, since mammalian Müller glia cells seem to have retained some of those regenerating abilities. Defining the mechanisms underlying the regenerative process in zebrafish may offer opportunities and new directions in the future of DR treatments [94].

There are established transgenic reporter lines for Müller glia as well, e.g., the *Tg(gfap:GFP)* line, in which Müller cells express the green fluorescence protein under the control of the *gfap* promotor. It was recently shown that larvae that are exposed to 4% glucose from 24 to 48 hpf show a significantly reduced number of Müller glial cells in the retina, which cannot be rescued post glucose exposure [95]. This reporter line has yet to be used in adult zebrafish models of diabetic retinopathy to evaluate its use in researching Müller glia activation or loss in diabetic conditions.

### 2.6. Photoreceptors/Neurodegeneration

Visual information is encoded in photoreceptors. In contrast to rodents, which are nocturnal animals and therefore have a rod dominated retina, zebrafish have a cone-dominated retina. Rods are highly sensitive and are most useful in dim-light conditions; therefore, they are mainly found in the peripheral parts of the human retina. The point of best visual acuity in the human retina, the fovea, which is the central part of the macula lutea, is mainly populated by cones, which are most active during daytime. Macular degeneration is one of the most important reasons for vision loss in humans. Even though the increased cone ratio in the zebrafish retina in comparison with the rodent retina makes the zebrafish an interesting model to study photoreceptor degeneration, it should be remembered that the fovea does not exist in the zebrafish retina (nor in other animal models). The effect of photoreceptor degeneration on the development of DR has only recently come into focus [96].

The hallmarks of diabetes-induced neuroglial degeneration include reactive gliosis (as discussed above), diminished neuronal function and neural-cell apoptosis. All of those have been observed to occur well before the first signs of microangiopathy in experimental models of DR or before the retina of diabetic human donors become visible [97,98]. Retinal ganglion cells and amacrine cells may be the first neuronal cells in which apoptosis becomes detectable, but photoreceptors also have an increased apoptotic rate [99]. The visible consequences of apoptotic cell death are reduced thickness of retinal layers, specifically the inner plexiform layer and the nerve fibre layer, as well as impaired ERG results due to photoreceptor degeneration and consequent dysfunction. Both of these effects can be observed in zebrafish models of DR. The data suggest that there are some discrepancies when analysing the effect of neurodegeneration in the zebrafish retina. Some studies found that the thickness of the inner plexiform layer significantly decreased, for example, in models of immersion-induced hyperglycaemia or STZ injection [16,47]. A decrease of nuclei in the outer nuclear layer in the genetic *pdx1* model was observed compared to age matched controls [53]. However, another study described an increase in retinal layer thickness in a model of immersion-induced hyperglycaemia [100], and it has even been shown that, in the *pdx1* model, there is an increase in nuclei in the inner nuclear layer [53]. These conflicting results clearly indicate that neurodegeneration in diabetic zebrafish models has not been sufficiently studied. A potential explanation could be that the regenerative capabilities of the zebrafish retina make it impossible to reliably quantify neurodegeneration, since the retinal layer thickness and the number of nuclei changes dynamically throughout the time of the experiment.

Photoreceptor degeneration has been widely and consistently observed in zebrafish, both morphologically [43,47,53] and functionally through abnormal ERG results [43,53,100]. This may be an interesting topic for further studies, with the new shift in research to consider the role that photoreceptors play in the pathogenesis of DR. 

## 3. Perspectives and Conclusions

Zebrafish have gained popularity as models for complications associated with diabetes and other metabolic diseases in recent years [101], with a wide array of different methods for inducing diabetic conditions published to this date [59]. Table 1 provides an overview of the various zebrafish models of DR published to date.

The focus of DR research in zebrafish rests on endothelial cell dysfunction and angiogenesis, since the ease of analysis of vasculature in transgenic lines is one of the main advantages of using zebrafish in research. Thus far, not much has been achieved in regard to the role of pericytes in zebrafish models of DR. One study found that, in the *pdx1* mutant, transgelin1 expression is reduced in mutants compared with age matched wildtype controls [53]. However, as mentioned above, this marker cannot reliably stain pericytes on zebrafish retinal vessels. With the discussed transgenic reporter lines for vascular mural cells in zebrafish, and the recent confirmation of the morphology and localisation of pericytes in zebrafish [80], this should change in the future. Future research should elucidate how pericytes behave in diabetic or other pathological conditions in zebrafish, revealing more about their function and potentially making them a target for further research. Apart from that, the role of inflammation in zebrafish models with impaired glucose metabolism has not yet been uncovered. This should also be taken into account for future investigations in the field, as there are multiple reporter lines for immune cells, including microglia [87].

In conclusion, zebrafish are no longer an exotic alternative model in diabetes research. Instead, research has focused on studying mechanisms and pathologies associated with hyperglycaemic conditions in zebrafish. Zebrafish demonstrate many advantages in such research. For example, hyperglycaemic conditions can be induced through easy and fast protocols, and the first effects on the hyaloid vasculature can be studied at larval stages. This is especially interesting when studying the effect of antiangiogenic drugs on the formation of new blood vessels. Adult zebrafish also show a reaction to hyperglycaemic conditions, with the *pdx1* mutants being the only known model organism in which retinal angiogenesis due to hyperglycaemia can be studied. Other important retinal phenotypes such as photoreceptor degeneration, increased vascular permeability, and the activation of Müller glial cells have been shown in various zebrafish models of DR, highlighting the similarities between mammalian and zebrafish models. Photoreceptor degeneration can be reliably modelled in zebrafish, including photoreceptor dysfunction, which can be quantified in an ERG. Furthermore, the zebrafish genome shows a high amount of shared genetic identity with humans. Through the possibility of inducing targeted mutations, this leads to an extensive number of possibilities for researchers in the field.

Important differences must also be considered. The ability of zebrafish to regenerate injured parts of the retina through the Müller glial cells is a relevant factor that differentiates zebrafish from mammals. Thus far, what role this regenerative capability plays in the development of DR-like pathologies is still unknown. Even though most of the anatomy of the retina and the individual cell types involved in the neurovascular unit are highly conserved, in contrast to the mammalian retina, zebrafish retinal vessels lay on top of the inner limiting membrane and do not form intraretinal plexuses. When choosing a zebrafish model of DR or analysing results of studies, researchers must be aware of these factors. In conclusion, zebrafish are a reliable model for various pathologies associated with DR, and they can be used to extend and improve the toolbox that mammalian models have provided for DR research so far.

## Figures and Tables

**Figure 1 cells-10-01313-f001:**
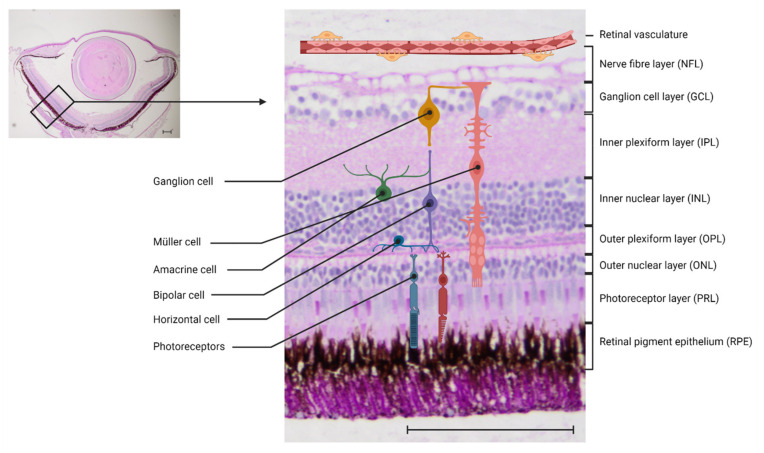
Anatomy of the zebrafish retina: Left: 4× magnification of a paraffin cut of the zebrafish retina, periodic-acid Schiff’s (PAS) stain. Right: 20× magnification of the zebrafish retina with a schematic overview of the different cell types. Scale bar = 100 μm.

**Figure 2 cells-10-01313-f002:**
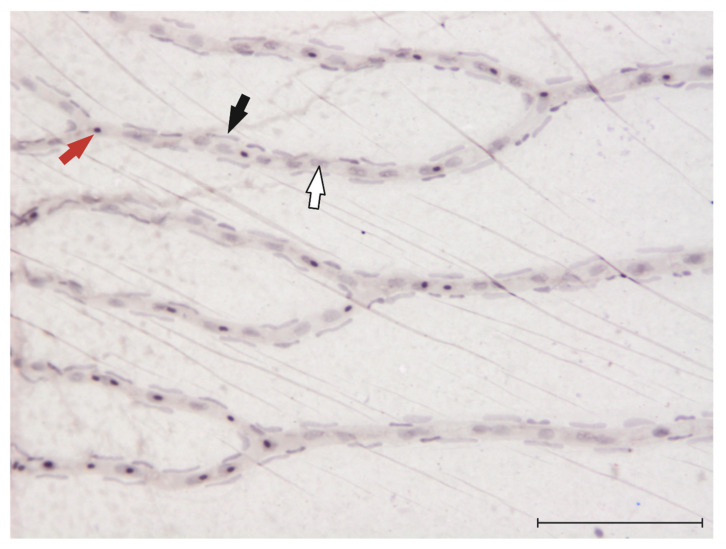
Visualization of endothelial cells, pericytes and erythrocytes in the zebrafish retina: 20× magnification of the zebrafish retina after digestion in 3% trypsin [81] and haemalum stain. Red arrow: erythrocyte, black arrow: pericyte, white arrow: endothelial cell. Scale bar = 100 μm.

**Table 1 cells-10-01313-t001:** Zebrafish models of diabetic retinopathy. Abbreviations: 4-HNE = 4-hydroxynonenal, dpf = days post fertilization, GCL = ganglion cell layer, GS = glutamine synthetase, hpf = hours post fertilization, i.p. = intraperitoneal, INL = inner nuclear layer, IPL = inner plexiform layer, MG = methylglyoxal, n.e. = not evaluated, ONL = outer nuclear layer, OPL = outer plexiforme layer, RGC = retinal ganglion cell, STZ = streptozotocin, “+” indicates positive findings.

Model	Induction	Angiogenesis	Endothelial Cell Dysfunction	Pericyte Loss	Müller Glia Activation	Photoreceptor Degeneration	Neurodegeneration
Gleeson, Connaughton et al. 2007 [16]	Exposure to alternating glucose/water solutions for 28 days (adult zf)	n.e.	n.e.	n.e.	n.e.	n.e.	+(decreased IPL thickness)
Cao, Jensen et al. 2008 [44]	Experimental hypoxia for up to 15 days (adult zf)	+	n.e.	n.e.	n.e.	n.e.	n.e.
Alvarez, Chen et al. 2010 [43]	Exposure to alternating glucose/water solutions for 30 days (adult zf)	n.e.	+ (thickening of vessel basement membrane, wider tight and adherens junctions)	n.e.	+	+(abnormal retinal histology, impaired cone ERGs)	-
Olsen, Sarras et al. 2010 [47]	i.p. or direct caudal fin injection of STZ (adult zf)	n.e.	n.e.	n.e.	n.e.	+(decreased PRL thickness)	+(decreased IPL thickness)
Carnovali, Luzi et al. 2016 [102]	Exposure to 4% glucose solution for 28 days (adult zf)	n.e.	+(increased vessel diameter, aneurysm-like structure, marked fragility of the anatomical structure)	n.e.	n.e.	n.e.	n.e.
Jung, Kim et al. 2016 [62]	Treatment with 130 mM glucose from 3–6 days post fertilisation (zf larvae)	-	+(increased vessel diameter, irregular and discontinuous staining of ZO-1)	n.e.	n.e.	n.e.	n.e.
Tanvir, Nelson et al. 2018 [100]	Exposure to alternating glucose/water solutions for 28 days (adult zf)	n.e.	n.e.	n.e.	n.e.	+(impaired ERG)	+(increased IPL and OPL thickness)
Ali, Mukawaya et al. 2019 [78]	Experimental hypoxia for up to 15 days (adult zf)	-(however: remodelling by intussusception)	+(decrease in ZO-1 abundance, increased vessel permeability)	n.e.	n.e.	n.e.	n.e.
Li, Zhao et al. 2019 [103]	Incubation with 500µM methylglyoxal with or without 30 mM glucose starting at 10 hpf to 4 dpf (zf larvae)	+(MG induces an increase in vascular area and branch points)	n.e.	n.e.	n.e.	n.e.	n.e.
Lodd, Wiggenhauser et al. 2019 [57]	CRISPR/Cas9 generated knockout zebrafish line for *glo1* + overfeeding of artemia (adult zf)	+	n.e.	n.e.	n.e.	n.e.	n.e.
Singh, Castillo et al. 2019 [95]	Exposure to 4 and 5% D-Glucose in a pulsatile manner from 3 hpf to 5 dpf (zf larvae, adult zf)	+(adult zf show an increased number of hyaloid blood vessel sprouts at 100 dpf after glucose treatment from 3 hpf to 5 dpf)	+(increased vessel permeability)	n.e.	(+) (reduced number of Müller glia cells)	n.e.	+(decreased IPL thickness, increased INL thickness, increased GCL thickness; decreased number of RGC)
Ali, Zang et al. 2020 [53]	*Pdx1* mutant fish generated through the Zebrafish Mutation Project (as described by Kimmel, Dobler et al. 2015) (adult zf)	+	+(vessel constriction and stenosis, reduction of average vessel diameter, reduced ZO-1 expression, reduced GLUT1 expression, increased vessel permeability)	(+) (reduced expression of transgelin1)	+(enhance expression of GS, hypertrophic changes)	+(reduced numbers of rods and cones, impaired ERG)	+(increased nuclei in the INL, decreased nuclei in the ONL)
Wiggenhauser, Qi et al. 2020 [55]	CRISPR/Cas9 generated knockout line for *pdx1* (zf larvae, adult zf)	+(at 6 dpf and in the adult retina)	+(increased number of endothelial cell nuclei, increased vessel permeability)	n.e.	n.e.	n.e.	n.e.
Lou, Boger et al. 2020 [56]	Incubation with 4-HNE (zebrafish larvae)	+(elevated vascular sprout formation)	+(increased branch diameters)	n.e.	n.e.	n.e.	n.e.

## Data Availability

Not applicable.

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
