# Peer review of "Advancing Diabetic Retinopathy Research: Analysis of the Neurovascular Unit in Zebrafish"

_cells, 2021, doi:10.3390/cells10061313_

Round 1

Reviewer 1 Report

as requested, I reviewed the manuscript (ID cells-1224038) "Advancing diabetic retinopathy research: Analysis of the neurovascular unit in zebrafish", by CS Middel, HP Hammes, J Kroll.

The review manuscript deals with the main research focusing on the zebrafish (Danio rerio) as model organism adopted in the study of diabetes and, more in detail, of diabetic retinopathy. Authors’ narrative approach is based on the description of the different cell types that form the neurovascular unit, and their physiological and pathophysiological roles in diabetic retinopathy. Moreover, a description of the specific advantages of the model is discussed.

In my opinion, the Introduction is quite effective and well-targeted, and the overall discussion contains all the elements for comprehension of the subject, for understanding the cited literature and for finding hints to deepen information.

The reading sounds like a good and comprehensible continuum, and there’s no evidence of fragmented information. The manuscript is quite conceived for giving the necessary keywords to enter the investigated subject, and actually reaches this goal.

On this basis, the author demonstrated scientific mastery of the research.

I only have a minor comment which might enhance the editorial appreciability, that is: I would rebalance the titles and therefore the number of the various paragraphs.

-           it is a bit of a stretch that there is a subsection 2.1 immediately after section 2; a few lines of general meaning are desirable before starting section 2.1.

-           section 3.3 should probably be section 3.

Thank you very much for your attention to my opinion.

Reviewer 2 Report

The paper by Dr. Middel describes zebrafish as diabetic retinopathy research model animal have some advantage in comparison with other animal such as mouse. The authors have some great review but some of description need to be clarified.

  1. Please clarify what sentences mention fig 1.
  2. Please describe function of glo1. (line 273)
  3. What is the staining in fig 2?
  4. Please clarify what fig 2 explain.
  5. What sentences explain table 1?
